# A Green Prospective for Learned Post-Processing in Sparse-View Tomographic Reconstruction

**DOI:** 10.3390/jimaging7080139

**Published:** 2021-08-07

**Authors:** Elena Morotti, Davide Evangelista, Elena Loli Piccolomini

**Affiliations:** 1Department of Political and Social Sciences, University of Bologna, 40126 Bologna, Italy; elena.morotti4@unibo.it; 2Department of Mathematics, University of Bologna, 40126 Bologna, Italy; davide.evangelista5@unibo.it; 3Department of Computer Science and Engineering, University of Bologna, 40126 Bologna, Italy

**Keywords:** green AI, sparse-views tomography, learned post-processing, CNN, UNet, tomographic reconstruction

## Abstract

Deep Learning is developing interesting tools that are of great interest for inverse imaging applications. In this work, we consider a medical imaging reconstruction task from subsampled measurements, which is an active research field where Convolutional Neural Networks have already revealed their great potential. However, the commonly used architectures are very deep and, hence, prone to overfitting and unfeasible for clinical usages. Inspired by the ideas of the green AI literature, we propose a shallow neural network to perform efficient Learned Post-Processing on images roughly reconstructed by the filtered backprojection algorithm. The results show that the proposed inexpensive network computes images of comparable (or even higher) quality in about one-fourth of time and is more robust than the widely used and very deep ResUNet for tomographic reconstructions from sparse-view protocols.

## 1. Introduction

Convolutional Neural Networks (CNNs), with their remarkable capacity of learning with multiple levels of abstraction, are giving new impetus to researchers working on inverse problems, and the imaging sector is one of the most involved field [1]. In fact, researchers have begun to tackle inverse imaging applications, such as denoising, deconvolution, in-painting, superresolution, and medical image reconstruction, with CNNs, and they all report significant improvements over state-of-the-art techniques, encompassing sparsity-based models derived from compressed sensing approaches [2,3].

In this paper, we focus on the X-ray Computed Tomography (CT) image reconstruction as a representative field of study of challenging inverse problem tasks for imaging. CT, in fact, is one of the most exploited diagnostic modalities in medical imaging, but the high radiation exposure per patient is unhealthy and may cause cancers. Hence, the definition and implementation of new safer protocols has become an active and interesting area of research in the inverse imaging community. In particular, the so-called Sparse-view CT (SpCT, or few-view CT) technique is a recent and popular proposal that lowers the radiation dose by reducing the number of X-ray projection views. In traditional CT (Figure 1a), about one thousand projections are executed over the 360-degree trajectory, whereas in the SpCT protocols (Figure 1b), the angular step between two adjacent scans is wider. Common sparse protocols consider scanning angular interval of one degree approximately. Furthermore, due to limitations of human anatomy or equipment manufactory, in special cases, the X-ray source may walk only a semi-circular or C-shape path (as depicted in Figure 1c), and the SpCT configuration is labeled as *limited angle CT*. Such low-dose tomographic approaches lead to incomplete CT projection data, and such subsampled measurements usually produce severe streaking artefacts on the Filtered Back-Projection (FBP) reconstructions. To address this, compressed sensing-based approaches have been investigated in the literature, minimizing the Total Variation (TV) or other sparsity-promoting priors combined with data fidelity terms [4,5,6,7,8,9,10,11]. Although the very accurate achievable results, the optimization approach has not been widely adopted yet in clinical setting because of its high computational cost.

As anticipated, the advent of deep learning is revolutionizing how researchers address CT (and, in particular, SpCT) image reconstruction, and a number of works have already been published trying to exploit the deep learning data-adaptivity for reaching high-quality medical images [12,13]. To this aim, we now focus on the paradigm sometimes referred to as *Learned Post-Processing* (LPP) or *Deep Artifact Correction*, which employs deep neural networks to suppress artefacts on roughly reconstructed images. This framework is graphically represented in Figure 2 for the specific context of SpCT, where the FBP algorithm is typically used to transform the subsampled sinogram data into the 2D medical image, and the LPP is performed at the end of whole reconstruction workflow to remove streaking artefacts and noise.

To the best of our knowledge, the first proposal of an LPP scheme for sparse-view CT dates back to 2016 with the pioneering paper [14], where chest images were restored by an end-to-end CNN that was pre-trained to learn the mapping between the FBP and artefact-free images. Later, many works have comprised UNet [15] architectures to fully take advantage of the down-sampling operations in the contracting path. In fact, since the FBP reconstruction from subsampled measurements are characterised by streaking (and hence global) artefacts, CNNs equipped with large receptive fields would better restore images from SpCT [16,17,18]. In addition, residual learning strategies have been embedded in the UNet architectures to preserve high texture details, which are important as well as difficult to recover during the expansive path [3,19,20]. Interestingly, the studies by Han et al. have already demonstrated the superiority of LPP strategies over some TV-based iterative algorithms for sparse-view CT reconstructions [17,21].

On the other hand, two main disadvantages of neural networks limit the effectiveness of the LPP approach. On one side, as highlighted in [22,23], the robustness of neural networks for medical applications is still a concern as they are vulnerable to unseen patterns. For instance, whenever the network takes as input an out-of-domain image, the noise- and artefact-free output may contain anatomical structures placed at wrong positions or even fake organ-like structures in the background. On the other side, the very deep structure of UNet requires a very expensive training in terms of time and consumed energy. To handle this constraint, the common choice adopted in the aforementioned papers consists of training the neural networks on small size bi-dimensional images: real medical 2D and 3D images are often too large for the present training possibilities.

Intertwined to these drawbacks, the green AI (Artificial Intelligence) line of thought is currently offering a new perspective and an interesting prospective that fit the inverse medical imaging community [24,25,26,27]. In fact, managing and reducing the energy cost of infrastructure and keeping a balance between model accuracy and sustainable computational costs, green AI is in line with medical requirements of real clinical settings.

### Aim and Contribution of the Paper

The aim of this paper is to propose a “green” (but nonetheless accurate and reliable) alternative to the widely used residual UNet scheme for the LPP reconstruction of CT images. Such choice may have many positive sides. First, looking for solutions that save time and energy is an essential prospective in our society. Secondly, lowering computational times can also reduce the cost of the hardware necessary to train the algorithms, making CT clinical exams and research more accessible. At last, due to the ongoing development of 3D CT imaging and the clinical requirement of almost real-time reconstructions, the forward pass in the LPP scheme must be as fast as possible.

In this scenario, the main contributions of this paper can be resumed as follows. On one side, we propose to use a very light convolutional network to correct artefacts on CT reconstructions from sparse views. The considered CNN allows for a very fast training, which can be adapted to large 2D images and 3D volumes. In particular, different from the UNet, the considered architecture is composed by only three inner layers and acts in single-scale modality on the input image. Due to its extreme light structure, it is expected not to overfit on the training set. On the other side, we validate the robustness and vulnerability of the proposed learned post-processing not only on a test set but even on out-of-domain cases, i.e., on images with slightly different patterns or statistics than the training samples. Such analysis is unusual in the literature, although it is well-known that it is important to investigate whether a neural network is vulnerable to perturbations on its input with respect to the training images to assess the CNN stability or overfitting.

The paper is organized as follows. In Section 2, we describe the LPP workflow for CT image reconstruction, and we illustrate the networks architecture; in Section 3, the numerical experiments are presented and discussed, and finally, Section 4 reports some conclusions.

## 2. Methods and Materials

In this section, we present and compare the two neural network architectures we tested for artefact removal on tomographic image reconstructions from sparse views. The first one is a residual UNet, labeled as ResUNet in the following. The second scheme is a very simple CNN composed by three layers and working in Single-Scale on the input image, and hence, it is denoted as 3L-SSNet.

As already mentioned, each proposed CNN is applied on the FBP reconstructed image to correct its artefacts (Figure 2). Formally, if we denote the artefact-corrupted image achieved by the FBP as *y* and the network output as $\bar{x}$, the Learned Post-Processing task can be formulated as:(1)$$\bar{x}=F_{\theta }\left(y\right)$$
where $F_{\theta }$ describes the neural network action on the input image *y* for the final restored image $\bar{x}$.

### 2.1. The ResUNet Architecture

State-of-art results in the image processing field have demonstrated that the popular UNet architecture by [15] operates efficiently whenever the input image shows global artefacts. As a matter of fact, it is known [28] that the pooling/unpooling strategy does permit to enlarge the receptive field of convolutional filters in such a way that it becomes possible to capture global information about the image in the lowest inner layers, whereas in the higher part, only the local information are processed. As a consequence, the UNet structure has been elected as the standard architecture even for sparse-view tomographic imaging tasks, where the streaking artefacts are visible on the whole image. In fact, the UNet scheme has been already successfully applied, working on the image or on a wavelet-based image transformation [29,30,31].

As observed in the theoretical work [21] by Han et al., in the case of sparse-view CT with FBP reconstruction, it can be proven that the residual manifold containing the artefacts is easier to learn than the true image manifold. In other words, it could be more effective to learn the residual map
(2)$$M_{R}:y⟼y+\bar{x}$$
than the correction map
(3)$$M_{C}:y⟼\bar{x}$$
for artefact suppression tasks. Hence, the image restoration model in Equation (Equation 1) turns into the following one:(4)$$\bar{x}=y+R_{\theta }\left(y\right)$$
which implies that the residual neural network $R_{\theta }\left(\cdot \right)$ must learn the artefacts manifold from *y*. Furthermore, residual deep networks have been already applied in the LPP step to remove artefacts from low-dose or sparse-view FBP reconstructions [17,18,32]. In this work, inspired by the network proposed in [33], we consider the residual-learning UNet architecture (labeled as *ResUNet* in the following) represented in Figure 3. In more detail, the ResUNet is a fully convolutional neural network with a symmetric encoder-decoder structure and pooling/unpooling operators to enlarge its receptive field. The pooling operations in the encoder naturally divide the network into distinct levels of resolution, which we will refer to as *l*, l=0,…,L, where L+1 is the total number of levels in the network. At each level, a fixed number $n_{l}$ of convolutional filters is applied, each one with the same number of channels $c_{l}$, which is constant along the level. Given a baseline number of convolutional channels $c_{0}$ (that corresponds to the number of channels in the first level), we will compute $c_{l}$ for the next levels with the recursive formula $c_{l+1}=2c_{l}$, l=0,…,L−1. In our specific implementation, we decided to fix L=4, $n_{0}=\cdots =n_{3}=3$ and $c_{0}=64$. As already said, the decoder is symmetric to the encoder, with upsampling layers instead of the pooling ones. Moreover, to maintain high-frequency information, skip connections are added between the last layer at each level of the encoder and the first layer at the correspondent level in the decoder. To lower the number of parameters with respect to the original architecture [33], we implement the skip connections as additions instead of the largely used concatenations.A residual connection is added between the input layer and the output layer too, following the implementation described in [3]. Each convolutional layer is composed by a Conv2D + BatchNormalization + ReLU structure, as it is common in the literature, except for the last layer, where we used a *tanh* activation function (as it is necessary to learn a residual map).

As intuitable, the ResUNet must learn a high number of parameters during its training.

### 2.2. The 3L-SSNet Architecture

Inspired by green AI ideas, we now consider a very simple architecture to learn the post-processing tasks. It is a three-layered fully Convolutional Neural Network with constant channel number equal to 128 and a filter size of dimension {9,5,3}. Each layer is the common Conv2D + BatchNormalization + ReLU block. A draft of the structure of the proposed network, denoted as 3L-SSNet in the following, is reported in Figure 4. As visible, the network does not contain pooling/unpooling steps; hence, it works in single-scale mode.

The 3L-SSNet architecture has been previously applied to post-process FBP reconstructions from low-dose CT in [34]. We remark that the geometry used by the authors in their experiments is very different from the one tackled in our study; hence, the network must learn different correction tasks in these works.

### 2.3. Receptive Field 

The portion of the input image *y* that is captured by each filter at a certain depth in the network is named the *receptive field*. Formally, the receptive field of a CNN at a fixed layer *t* is defined as the portion of the input image *y* that produces a certain pixel of the feature map at the *t*-th layer [35,36]. Since we are interested in comparing our neural network architectures in terms of their receptive field, we need to derive a formula to compute it for a given network.

For each layer *t*, let $k_{t}$ and $s_{t}$ be its kernel dimension and the stride, respectively. Moreover, let $r_{t}$ be the receptive field; the receptive field of the input layer is $r_{0}=1$. The value of $r_{t}$ can be computed with the recursive formula [35]:(5)$$\begin{array}{c} r_{t}=r_{t-1}+A_{t} \end{array}$$
where $A_{t}$ is the non-overlapping area between subsequent filter applications. Note that $A_{t}$ can be simply computed as
(6)$$\begin{array}{c} A_{t}=\left(k_{t}-1\right)\prod_{i=1}^{t}s_{i} \end{array}$$
which implies that the receptive field at each *t*-th layer is:(7)$$\begin{array}{c} \left\{ \begin{array}{cc} \; & r_{0}=1\\ \; & r_{t}=r_{t-1}+\left(k_{t}-1\right)\prod_{i=1}^{t}s_{i} \end{array}\right. \end{array}$$

Equation (Equation 7) shows that the receptive field scales linearly with the depth of the network if the kernel dimension is fixed, while it is exponentially related with the strides. For this reason, reducing the image dimension with pooling operators while processing the image exponentially enlarges the receptive field of the convolutions.

As reported in Figure 3, ResUNet has the maximum receptive field of 172×172 pixels, corresponding to the 11.28% of the input image. On the contrary, the 3L-SSNet is significantly smaller since its receptive field is at most 15×15 pixels in its last layer (Figure 4).

### 2.4. Training of the Networks

To train the networks, we have used numerical simulations generated from full-sampling CT images provided by the AAPM Low Dose CT Grand Challenge [37]. The downloaded images are 512×512 pixels and are chest reconstructions from full-dose acquisition data; thus, we used them as ground truth images after scaling them in the interval [0,1]. We used ten patients (3306 images) for the training phase and one patient (357 images) for testing. As visible from Figure 5 where a slice of the test patient is reported in its ground-truth (GT) original version, the considered samples are not noise- nor artefact-free at all. We remark that this feature may lead to some little corruptions on the CNN restored images.

Given the training set $D=\{{\left(y_{i},x_{i}^{GT}\right\}}_{i=1,\ldots ,N_{D}}$, where $y_{i}$ are the input samples to the network, and $x_{i}^{GT}$ are the correct labels, we train the parameters θ such that if ${\bar{x}}_{i}=F_{\theta }\left(y_{i}\right)$ is the restored image given $y_{i}$, we have
(8)$$\theta^{*}=\arg\min_{\theta }\frac{1}{N}\sum_{i=1}^{N}\ell \left({\bar{x}}_{i},x_{i}^{GT}\right).$$

In our implementation, $\ell \left({\bar{x}}_{i},x_{i}^{GT}\right)=\left|\right|{\bar{x}}_{i}-x_{i}^{GT}{\left|\right|}_{2}^{2}$. In ResUNet, training is performed by running Stochastic Gradient Descent (SGD) for 50 epochs with a batch size equal to 8 and Nesterov Acceleration with momentum equal to 0.99. The step size for SGD decreases with polynomial decay, going from $10^{-2}$ to $10^{-5}$ during training. To increase the stability over the first iterations, we clipped the gradient to 5.

In 3L-CNN, the training parameters are exactly the same as for ResUNet, except for the fact that we ran Adam instead of SGD, as we noticed that in that situation, SGD got stuck in a local minimum after a bunch of iterations. The training was performed on two Nvidia GeForce RTX 2080Ti (NVIDIA, Santa Clara, CA, USA).

### 2.5. Network Comparison 

To complete the comparison between the considered ResUNet and 3L-SSNet, we report in Table 1 further useful details. Focusing on the number of parameters and the seconds for training of each structure, we observe that 3L-SSNet has only 85,000 parameters, and it requires less than one minute to complete an epoch (corresponding to a quarter of the ResUNet time). The Green AI FLOPs index reflects the faster performance of the 3L-SSNet even in the forward execution to process new images in real-time. The higher computational advantage of the 3L-SSNet network is clearly visible.

## 3. Experimental Results and Discussion

In this section, we report and discuss the representative experiments performed to test the effectiveness of the considered networks.

We developed our workflow in Python, and the code is available at https://github.com/loibo/3LSSNet.

To build the training and testing data sets, we computed the synthetic projection data using the ASTRA toolbox [38], providing routines for the forward 2D projections of the ground truth images. To simulate the sparse-view geometry, we considered two different protocols: a full angular acquisition with 1-degree spaced projections (denoted as *full-range* in the following) and a reduced scanning trajectory limited to 180 degrees with 180 projections (denoted as *half-range* in the following). We added to the sinograms white Gaussian noise with $10^{-2}$ noise level, and finally, we computed the FBP reconstruction by ASTRA routine.

### 3.1. Metrics for Image Quality Assessment

To evaluate the quality of reconstructed images quantitatively, we consider the following widely used metrics. Given a reconstructed image *x* of *n* pixels, we compute its relative error (*RE*)
(9)$$RE=\frac{∥x-x^{GT}{∥}_{2}^{2}}{∥x^{GT}{∥}_{2}^{2}}$$
and the Peak Signal-to-Noise Ratio (*PSNR*) index
(10)$$PSNR=20\log_{10}(\frac{n\cdot max(x^{GT})}{∥x-x^{GT}{∥}_{2}}).$$

To better evaluate the visual appearance of an image, we also compute the well-known Structural Similarity (SSIM) index [39], measuring the perceptual difference between two similar images, and the Feature Similarity (FSIM) index [40], which should better interpret the low-level features, conveying the most crucial information according to the human visual system. We remark that $x^{GT}$ has values in [0,1], whereas each output image *x* is visualized in its proper interval $\left[x_{min},x_{max}\right]$.

### 3.2. Results on the Test Set

In this paragraph, we discuss the results obtained on the test set. We analyse, in particular, the reconstructions of the slice in Figure 5a, considering the projections acquired with both the full-range and half-range geometries described above. We compare the results computed by the FBP algorithm and the LPP images with ResUNet and 3L-SSNet networks.

In Table 2 and Table 3, we report the average values of the considered metrics in the full-range and half-range cases, respectively. We first remark the very poor values achieved by the FBP, which are motivated by its difficulty in recovering the actual intensities of the ground truth images. Nevertheless, both the LPP images enhance such quality indices significantly. The 3L-SSNet performs better with full-range geometry, whereas for half-range, the ResUNet is outperforming.

In Figure 6 (full-range geometry) and Figure 7 (half-range geometry), we focus our visual inspection on the reconstructions of the slice in Figure 5a. From the crops of Figure 6, we observe that the images learned by the two networks look similar. The streaking artefacts of the FBP reconstruction (Figure 6d) are not completely removed in either Figure 6e,f. The area shown in Figure 6g is mainly corrupted by noise, which is cleaned well, especially in the reconstruction with ResUNet (Figure 6h).

In Figure 7, we depict the reconstructions obtained with the half-range geometry. In this case, the 3L-SSNet network produces more accurate images. In Figure 7f, the streaking artefacts are less visible than in Figure 7e. Moreover, the low contrast objects (pointed by the arrows) are more clearly distinguishable and have sharper contours in Figure 7i than in Figure 7h.

### 3.3. Tests on Out-of-Domain Data

It is well known that one critical drawback of neural networks is their performance on unseen data; hence, we now test the considered networks on *out-of-domain* data. We apply the algorithms to two different projection sets: the first one is from the patient test data with increased noise with respect to the training set (Section 3.3.1); the second one is obtained from a digital image of the XCAT phantom [41], used elsewhere in the literature to test neural networks on X-ray images [42] (Section 3.3.2). In this case, the test problem has been built as for the training set.

#### 3.3.1. Test on Unseen Noise

We analyse the results of the algorithms on the test simulations, obtained by adding Gaussian noise with level $2\times 10^{-2}$ to the projections of the ground truth images.

In Table 4, we report the the metrics computed on our reference slice for both the geometries. We observe that with full-range geometry the 3L-SSNet performs better, whereas the ResUNet shows superior values in the half-range case.

However, even in this case, the visual inspection is not fully consistent with the metrics since the reconstruction obtained by 3L-SSNet with half-range geometry shows the highest quality. In Figure 8 (half-range geometry), the image in Figure 8b learned by 3L-SSNet is less noisy than the crop in Figure 8a from ResUNet; in the second zoom, the low contrast objects pointed by the arrow are far more contrasted in the 3L-SSNet reconstruction in Figure 8c than in the ResUNet (Figure 8d). Moreover, a noisy pattern is still visible inside the dark background of the lungs in Figure 8c, reflecting the difficulty of the residual network in handling unseen noise.

#### 3.3.2. Test on Unseen Image

At last, we analyse the LPP reconstructions of the XCAT digital image, which does not belong to the AAPM Low Dose CT Grand Challenge data set. The ground truth image is displayed in Figure 9a, together with two zooms-in in Figure 9e,i. We observe that it has different features with respect to training images since it is completely noise-free and is constituted by flat regions containing small sharp objects of interest.

We do not report the images obtained in the full-range case, where the 3L-SSNet metrics (RE = 0.0506, SSIM = 0.9213) outperform the ResUNet (RE = 0.0567, SSIM = 0.8503), even if the final images have been restored well visually and look very similar. We focus our analysis on the half-range case, whose results are depicted in Figure 9 and are much more significant. In the first crop (Figure 9e–h), it is evident that the noise is better suppressed by the 3L-SSNet, which gives images with more uniform areas (hence, more similar to the GT one). Concerning the second zoom-in (Figure 9i–l), the artefact (pointed by the arrow in Figure 9k), which was surprisingly introduced in the ResUNet reconstruction, catches our attention. The darker contour following the border of the chest is not present in the Ground Truth or the FBP reconstruction.

### 3.4. Discussion

Our numerical results demonstrate the potential of both the ResUNet and the 3L-SSNet in correcting the FBP reconstructions, which are affected by severe corrupting effects and lack of contrast. In particular, the two networks provide comparable results in terms of metrics and image quality when applied on test data coherent to the training samples. The comparison between Table 2 and Table 3 highlights the ResUNet superiority in the case of half-range geometry, where less projections are available, but the visual inspection of Figure 6 and Figure 7 reveals very similar reconstructions in all the shown images and zooms.

The artefact correction by ResUNet becomes less effective than 3L-SSNet when processing out-of-domain data, i.e., when the input images are characterized by features different from those learned from the training samples, such as the image dealt with in Section 3.3. It reflects the trend of very deep neural networks to overfit on the learned image patterns.

In general, even if the receptive field of the 3L-SSNet is extremely smaller than the ResUNet one, its 15 × 15 RF area is big enough to discern the SpCT artefacts (due to the FBP reconstruction) from the specific patterns of the ground truth images. We think that this could explain why the 3L-SSNet post-processing has comparable effects to the ResUNet ones.

## 4. Conclusions

In this paper, we propose 3L-SSNet, a non-intensive computation neural network for a Learned Post-Processing reconstruction algorithm in CT. The proposal fits with the Green AI research, studying computationally cheap algorithms to save energy and be inclusive. Moreover, in the tomographic setting, reducing time as much as possible is important to make the algorithms usable in clinics.

The results obtained by 3L-SSNet on in-domain images (i.e., test images coherent to the ones used for the network training) are comparable to the output of ResUNet, a widely used very deep architecture, in terms of metrics and visual inspection.

We also tested both networks on out-of-domain images (i.e., CT images not belonging to the training nor test set), and we surprisingly got reconstructions from 3L-SSNet sometimes more accurate than the ones by ResUNet. The deep ResUNet, besides requiring computational time four times greater than 3L-SSNet, does not handle the unseen features efficiently.

Motivated by these very good results, we intend to test other Green networks for possibly reducing CT artefacts in different reconstruction frameworks. Moreover, a 3L-SSNet shallow-like network can be tested for artefact correction in other inverse problems in imaging, such as deblurring or super resolution.

## Figures and Tables

**Figure 1 jimaging-07-00139-f001:**
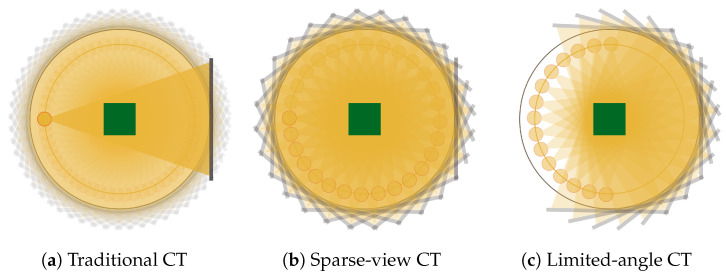
The three different CT geometric protocols. In the traditional setting (**a**), the X-ray source and the detector walk a full circle trajectory with a very small angular step, which is enlarged in sparse-view CT (**b**), whereas in limited-angle CT (**c**), the source-detector rotation is restricted to a C-shape path.

**Figure 2 jimaging-07-00139-f002:**

Graphical draft of the considered two-step workflow for tomographic reconstruction from sparse-view data.

**Figure 3 jimaging-07-00139-f003:**
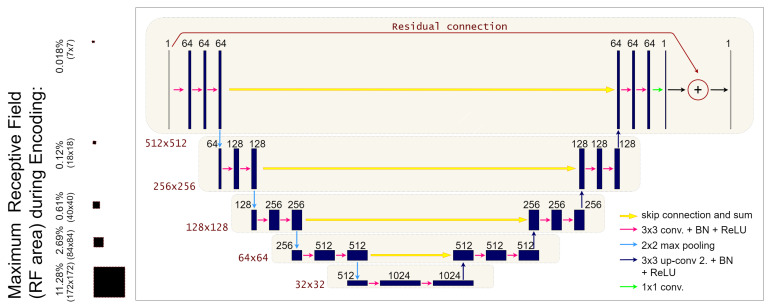
On the right: graphical representation of the ResUNet architecture; On the left: details on the maximum receptive fields for each of the five levels of the network encoder (RF percentage respect to the input 512 × 512 image and size of RF).

**Figure 4 jimaging-07-00139-f004:**
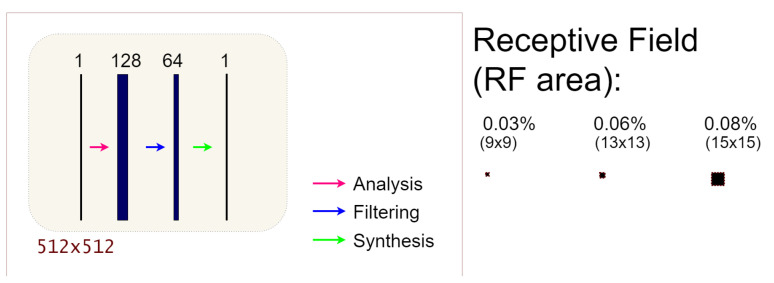
On the **left**: graphical representation of the 3L-SSNet architecture; on the **right**: details on the receptive fields for each of the three layers of the network (RF percentage respect to the input 512 × 512 image and size of RF). The name of the three layers follows the notation in [34].

**Figure 5 jimaging-07-00139-f005:**
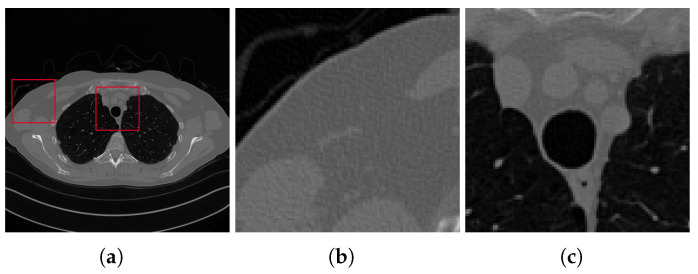
Ground truth image (**a**) and the two considered zooms-in (**b**,**c**), which are depicted by the red squares on the full image (**a**).

**Figure 6 jimaging-07-00139-f006:**
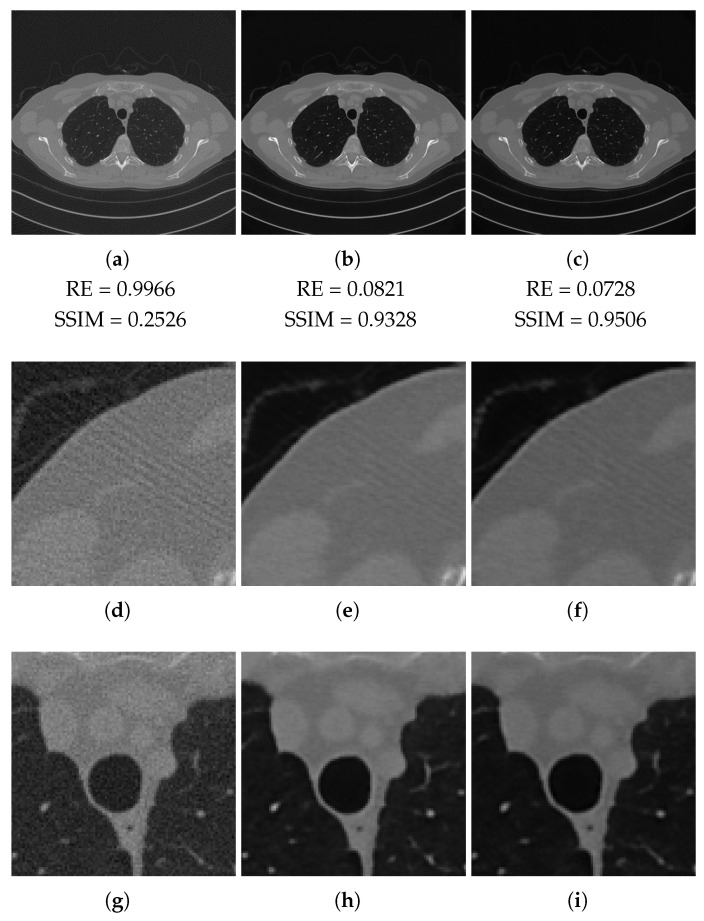
Full-range geometry reconstructions. The results obtained with FPB (**left column**), ResUNet (**central column**) and 3l-SSNet (**right column**). Below each image, the values of its RE and SSIM metrics.

**Figure 7 jimaging-07-00139-f007:**
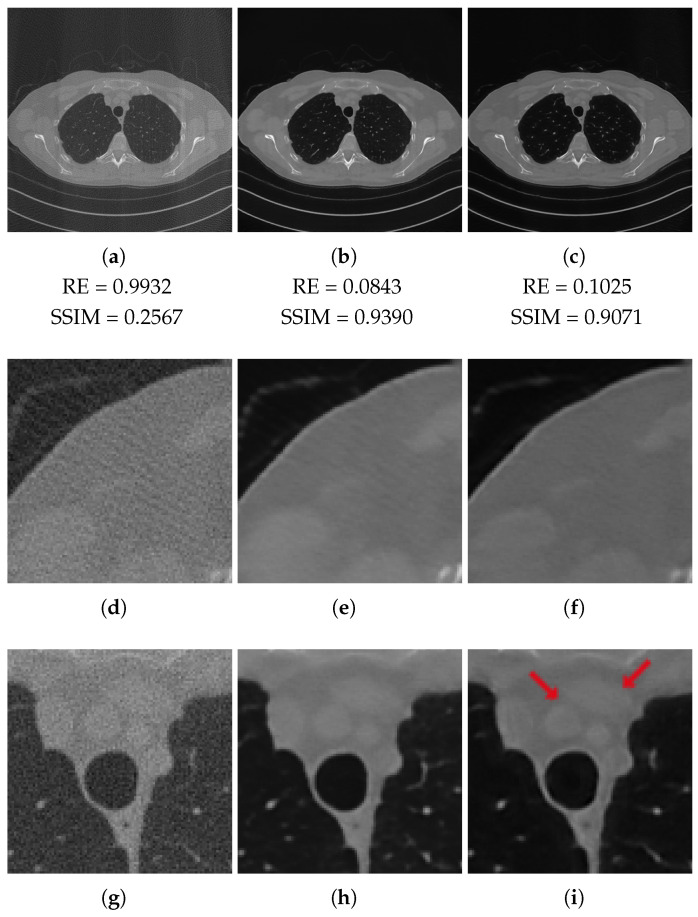
Half-range geometry reconstructions. The results obtained with FPB (**left column**), ResUNet (**central column**) and 3L-SSNet (**right column**). Below each image, the values of its RE and SSIM metrics.

**Figure 8 jimaging-07-00139-f008:**
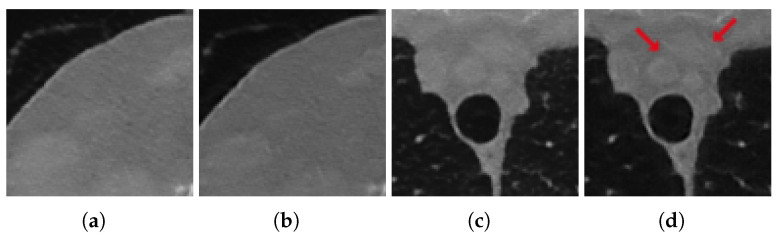
Crops of the reconstructions of the test patient with unseen noise and half-range geometry. ResUNet in (**a**,**c**), 3L-SSNet in (**b**,**d**).

**Figure 9 jimaging-07-00139-f009:**
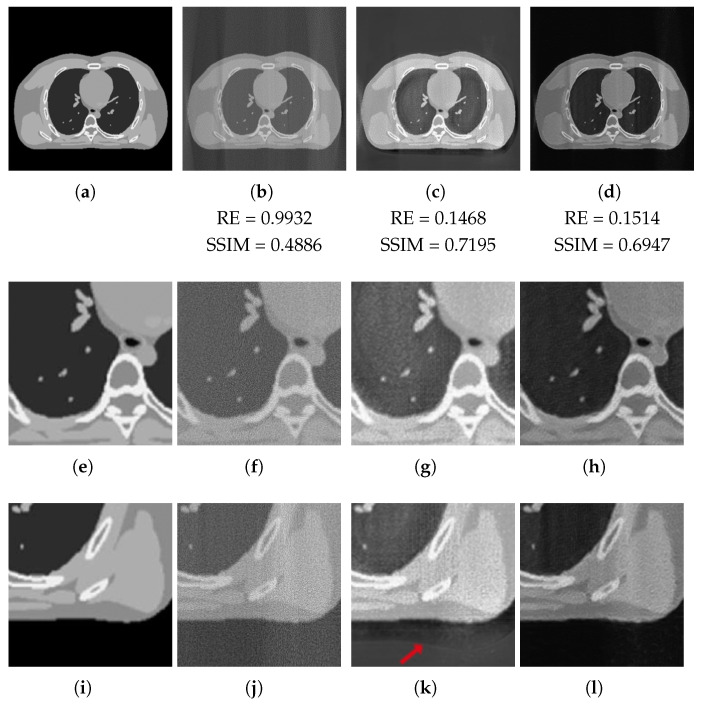
XCAT phantom test image with half-range geometry. From the left to right: (**first column**): Ground Truth image (**a**) and the considered zooms-in (**e**,**i**). Reconstructions from half-range geometry with FBP (**second column**), ResUNet (**third column**) and 3L-SSNet (**fourth column**).

**Table 1 jimaging-07-00139-t001:** A comparison of the cost of the considered networks. The training time is expressed in sec/epoch in the third column.

	Parameters	FLOPs	Training Time
ResUNet	$34.5\times 10^{6}$	$406\times 10^{9}$	209
3L-SSNet	$85\times 10^{3}$	$44\times 10^{9}$	53

**Table 2 jimaging-07-00139-t002:** The average of the full-reference metrics on the test set in the case of full-range geometry.

	RE	PSNR	SSIM	FSIM
FBP	0.9966	86.42 (33.89)	0.2924	0.5456
ResUNet	0.0942	106.99 (41.95)	0.9262	0.9709
3L-SSNet	0.0840	107.92 (42.32)	0.9480	0.9627

**Table 3 jimaging-07-00139-t003:** The average of the full-reference metrics on the test set in the case of half-range geometry.

	RE	PSNR	SSIM	FSIM
FBP	0.9932	86.45 (33.90)	0.2962	0.6819
ResUNet	0.1016	106.38 (41.71)	0.9324	0.9478
3L-SSNet	0.1309	104.34 (40.91)	0.9021	0.9474

**Table 4 jimaging-07-00139-t004:** Full-reference metrics on the test image with unseen noise in full-range and half-range cases.

	FBP		ResUNet		3L-SSNet	
	RE	SSIM	RE	SSIM	RE	SSIM
Full-range	0.9966	0.2526	0.0966	0.9172	0.0896	0.9295
Half-range	0.9932	0.2567	0.0986	0.9212	0.1162	0.8866

## Data Availability

Public data set AAPM Low Dose CT Grand Challenge available at https://www.aapm.org/grandchallenge/lowdosect/.

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
