# Peer review of "A Green Prospective for Learned Post-Processing in Sparse-View Tomographic Reconstruction"

_2313-433X, 2021, doi:10.3390/jimaging7080139_

Round 1
Reviewer 1 Report
In this paper, the authors propose a new neural work architecture, a 3-layer convolutional neural network, to post-process the filtered back-projection reconstruction of a CT scan. Comparing to a deep neural network like ResUNet, the authors demonstrate that the proposed CNN is more robust, provides similar results, and consumes a lot less energy.
This paper is well-organized. However, my main concern is about the novelty and usefulness of this method.
- In the following paper, these authors reconstruct the limited-angle CT scan from a simple 3-layer CNN, without accessing the FBP reconstructed results. It is even less energy-savvy compare to your architecture. Please comment.
Chen, Hu, et al. "Low-dose CT via convolutional neural network." Biomedical optics express 8.2 (2017): 679-694.
- In the following paper, these authors also use light CNN with FBP, but in a different way. How would you compare FBP combined with CNN, versus FBP followed by CNN as two independent steps?
Yamaev, Andrei, et al. "Lightweight denoising filtering neural network for FBP algorithm." Thirteenth International Conference on Machine Vision. Vol. 11605. International Society for Optics and Photonics, 2021.
My other questions are:
- Section 3.3.1. Test on unseen noise. Could you provide some metrics? The difference is not obvious in the image
- Section 3.3.2. Test on unseen image. The unseen image is still very similar to the training image. Could you test on some images that are more different, for example, brain image and vessel image?
- The repository in your Github is empty https://github.com/loibo/3LSSNet.
- Figure 6. no caption for figure d-I, do you miss some boxes to denote the area?
Author Response
we uploaded a pdf file

Reviewer 2 Report
The authors tackle the problem of removing artifacts from Filtered Back-Projection (FBP) reconstructions using neural network.
While this idea has already been explored, the main contribution is to replace the usually heavy network by a lighter neural network architecture (85000 parameters). They validate their proposal on a convincing dataset for full-range and half-range geometries in X-ray Computed Tomography reconstruction.
They show that their simple architecture, denoted 3L-SSNet, performs similarly to a heavy residual neural network.
The second contribution is to show that their trained network is stable i) to noise that is twice the noise level used in the training phase, ii) to one other image form another dataset.
Some general remarks:
- Figure 1: you should add a more complete description.
- Section 2.1: the idea of using a residual neural network with an auto-encoder to correct FBP reconstructions is not new. You could clarify that and give some references.
- There is no code available in the Github web page. I believe this is really important to make it public. It will also help us to better review this article.
- Section 3.3.2, why don't you report the images in the full-range case? Add at least one sentence precising if the visual conclusion are similar to the half-range case or not.
Some typographical mistakes:
- line 25: missing point.
- line 235: 'motivated by the its difficulty'.
- line 237: 3L-SSn is worst in term of FSIM, not SSIM.
- line 313: 'We alsoi tested'.
- line 321: 'such as deblur ' -> deblurring.
Author Response
we upload a pdf file

Round 2
Reviewer 1 Report
Thank the authors for revising the manuscript. All my questions are properly addressed.